# Fatty Acids Composition of Pasture Grass, Yak Milk and Yak Ghee from the Four Altitudes of Qinghai–Tibet Plateau: A Predictive Modelling Approach to Evaluate the Correlation among Altitude, Pasture Grass, Yak Milk and Yak Ghee

**DOI:** 10.3390/ani14202975

**Published:** 2024-10-15

**Authors:** Runze Wang, Jinfen Yang, Binqiang Bai, Muhammad Irfan Malik, Yayu Huang, Yingkui Yang, Shujie Liu, Xuefeng Han, Lizhuang Hao

**Affiliations:** 1Key Laboratory of Plateau Grazing Animal Nutrition and Feed Science of Qinghai Province, Qinghai University, Xining 810016, China; runzewang2000@foxmail.com (R.W.); 15500741942@163.com (J.Y.); binqiangbai@163.com (B.B.); yykui@qhu.edu.cn (Y.Y.); mkylshj@126.com (S.L.); 2Department of Veterinary Sciences, University of Turin, Largo Braccini 2, 10095 Grugliasco, Italy; dr.irfan279@gmail.com; 3PEGASE, INRAE, Institut Agro, 35590 Saint-Gilles, France; 4CAS Key Laboratory for Agro-Ecological Processes in Subtropical Region, Hunan Provincial Engineering Research Center for Healthy Livestock and Poultry Production, Institute of Subtropical Agriculture, The Chinese Academy of Sciences (CAS), Changsha 410125, China

**Keywords:** Qinghai–Tibet Plateau, fatty acids composition, correlation analysis, yak, altitude

## Abstract

This study examines how altitude affects the fatty acid composition of pasture grass, yak milk, and yak ghee on the Qinghai–Tibet Plateau. It aims to understand how environmental factors, particularly altitude, influence the nutritional quality of these products. The results indicate that higher altitudes correlate with a greater presence of beneficial unsaturated fatty acids, such as oleic acid (C18:1) and linoleic acid (C18:2n6c), suggesting that yaks adapt to their high-altitude surroundings. These adaptations positively impact the quality of yak milk and yak ghee, essential food sources for local communities. By emphasizing the connection between altitude and fatty acid profiles, this research offers insights that can enhance dairy production practices. Ultimately, it aims to improve nutrition and food security for residents in high-altitude regions, ensuring access to nutritious food options.

## 1. Introduction

Yaks are remarkable animals adapted to harsh environments, primarily found in the Qinghai–Tibet Plateau region, that have unique genetic potential and physiological traits that allow them to thrive in high-altitude, low-oxygen environments [1]. Over 90% of the world’s yaks are found in China, where they are adapted to extreme environments and provide livestock products for plateau inhabitants [2,3]. Yaks are vital for the livelihoods and food security of residents, providing high-quality meat and fur for human consumption, as well as highly nutritious milk. Additionally, yak milk is processed to produce yak ghee [4]. The residents of these highlands have been utilizing yak ghee since ancient times as food and medicine to heal wounds, and they also use it to worship the gods due to its purity. Additionally, the importance of yak ghee has increased due to limited supply or the unavailability of plant-based oils, making it the primary oil utilized in their daily diet [5].

Yak milk, often referred to as natural concentrated milk [6], is characterized by a high solid content (16.3–19.0%), fat content (5.6–8.8%), and protein content ranging from 4.68 to 5.41% [7]. The composition of milk is influenced by various environmental factors, including the breeding area, climate, breeding season, pasture grass growth stage, and pasture grass quality [8,9]. The composition of yak milk, as the raw material for yak ghee, directly affects the composition and quality of the yak ghee [10]. Yak ghee, a traditional handmade dairy product common in the Qinghai–Tibet Plateau, is a key ingredient in Tibetan cuisine and an essential energy source for plateau residents to combat cold temperatures [11,12]. It is bright yellow in color, has a high fat (87.7%) content and is produced through a natural fermentation process [13]. Consuming yak ghee can reduce the need for additional unsaturated fatty acid (UFA) and offers health benefits, including anti-cancer, anti-atherosclerosis, and osteoporosis inhibition, primarily due to its high conjugated linoleic acid (CLA) content [14,15]. Compared to cow ghee, yak ghee contains higher levels of UFA, making it a significant source of CLA [4].

Yak milk fatty acids differ based on pasture grass composition across altitudes, leading to variability in yak ghee. Studies have documented that the fatty acid profiles of yak ghee vary in high-altitude regions of Nepal [16]. Additionally, seasonal and altitude-based variations in yak milk have been noted on the Qinghai–Tibet Plateau [12,17]. However, there is currently a lack of research on the composition of yak ghee from different altitude areas on the Qinghai–Tibet Plateau. Additionally, studies on how pasture grass composition at different altitudes affects the nutritional content of yak milk and yak ghee are also deficient.

This study examines how fatty acid composition varies with altitude in the pasture grass–yak milk–yak ghee system and within a single altitude. The aim is to create predictive models, explore unique fatty acid biosynthesis in high-altitude yaks, and assess the impact on milk and ghee quality. The results will reveal how environmental factors influence the health benefits of yak products for plateau residents.

## 2. Materials and Methods

### 2.1. Sample Collection

The experiment was carried out from August to September on the Qinghai–Tibet Plateau, with an average temperature of 10 °C and humidity at 45%. Twenty-two counties across six regions in the Northern Tibetan Plateau (Yushu, Guoluo, Hainan, Huangnan, Haixi, and Haibei) were selected. The sampling points were divided into four groups, A1-A4, according to the altitude gradient of every 500 m. Samples of pasture grass, milk, and ghee were collected at each sampling point, with a biological replicate number of 10 for each type of sample. Pasture grass samples were collected from each pastoralist’s own pasture using several quadrats (1 m × 1 m). Collected pasture grass samples were processed to remove unnecessary materials such as stones, wood, and inedible objects. The cleaned samples were then placed on ice and immediately transported for storage at −20 °C for further analysis.

In the area where pasture grass was collected, six yaks from the herds of local herdsmen were also selected for the collection of yak milk, resulting in a total of 15 samples (50 mL) of mixed yak milk from each location. Before morning grazing, the milking process involved allowing the calves to stimulate milk letdown in their mothers, followed by manually collecting milk samples through hand milking by an experienced herder. After milking, samples were placed into centrifuge tubes. Additionally, yak ghee made by pastoralists themselves was collected using double-headed stainless-steel spoons and stored in falcon tubes of 15 mL. All collected yak milk and yak ghee samples were cooled in liquid nitrogen and transported to a laboratory, where they were frozen at −20 °C for storage. The distribution of the sampling points is shown in Figure 1.

### 2.2. Sample Analysis

The fat content in pasture grass was determined using an XT15i automatic fat analyzer (ANKOM XT15i, ANKOM, Macedon, New York, USA). The fat content in yak ghee and yak milk was determined according to standards (AOAC, 1999) [18]. A 10 mL sample of ghee or milk was mixed with 1.5 mL of concentrated ammonia solution, followed by 10 mL of ethanol, 10 mL of petroleum ether, and 10 mL of ether in a separation funnel. The upper layer was then collected, and the solvents (ether and petroleum ether) were removed by heating and subsequently weighed. The fatty acid composition of the pasture grass samples was determined by extracting total lipids with a chloroform/methanol (1:1) mixture, including an internal standard (23:0), followed by transesterification with methanolic HCl. The resulting fatty acid methyl esters were purified using thin layer chromatography with a hexane/diethyl ether/acetic acid (85:15:1) solvent system and analyzed using gas chromatography [19]. The fatty acid composition of yak milk and yak ghee was evaluated according to Or-Rashid et al. [20]. A 0.2 g sample was placed in a 2 mL glass centrifuge tube, and 1 mL of chloroform–methanol solution was added and sonicated for 30 min. The supernatant was esterified with 2 mL of 1% sulfuric acid-methanol solution and placed in a water bath at 80 °C for 30 min. Then, the mixture was extracted with 1 mL of n-hexane and washed with 5 mL of pure water. The 25 μL of methyl salicylate was regarded as an internal standard. Later, 500 μL of the supernatant was mixed and added to the sample bottle for GC detection with an injection volume of 1 μL and a split ratio of 10:1. The separation was performed in a gas chromatograph (GC) (Agilent) on a capillary column (100 m × 0.25 mm × 0.25 μm, Supelco SP-2560). The temperature program was set as follows: the initial temperature was maintained at 70 °C and later increased at a rate of 8 °C/min to 110 °C which was maintained for 2 min. Again, the temperature was raised at a rate of 5 °C/min to 170 °C for 10 min. Finally, the temperature was increased at a rate of 4 °C/min to 240 °C for 45 min. The carrier gas was helium with a flow rate of 1.0 mL/min. A QC sample was set when the same number of experimental samples were separated every time in the sample queue to detect and evaluate the stability and repeatability of the system. Fatty acids were identified and quantified based on the results of the gas chromatography-mass spectrometry analysis.

### 2.3. Data Analysis

Experimental data were organized and edited using Excel 2019 software. Subsequently, multivariate analysis of variance (MANOVA) was performed using RStudio statistical analysis software (version 4.4.1) to validate whether fatty acids in pasture grass–yak milk–yak ghee systems at different altitudes showed significant changes (*p* < 0.05). Additionally, linear discriminant analysis was conducted. The overall approach of this experiment is shown in Figure 2.

## 3. Results and Discussion

### 3.1. Fatty Acid Profiles in Pasture Grass–Yak Milk–Yak Ghee System

Table 1 and Table 2 present the fatty acid compositions of pasture grass and yak milk. In pasture grass, lauric acid (C12:0) was significantly higher (*p* < 0.001) at A2 altitude as compared to A1, A3, and A4. Similarly, myristic acid (C14:0) was significantly higher in A2 as compared to A1, A3, and A4. Stearic acid (C18) was significantly higher (*p* < 0.001) at the A1, A2 and A3 altitudes as compared to the C4 altitude. C18:2n6c was significantly higher (*p* < 0.001) in A4 and A1 as compared to A2 and A3. Linolenic acid and eicosadienoic acid change significantly with altitude, whereas there are no significant differences in stearic acid, arachidic acid, and specific polyunsaturated fatty acids, which is similar to results from pasture grass collected at similar altitudinal gradients [17]. It is also observed that an increase in altitude influences the content of long-chain fatty acids in pasture grass, such as C20:2 and linoleic acid, a phenomenon confirmed in previous studies. The fatty acid content in pasture grass is affected by several factors, including the type of pasture grass, growth stage, fertilization, and preservation methods [21]. These changes may relate to the plant’s adaptation mechanisms to high-altitude conditions [22,23].

In yak milk, palmitic acid (C16:0) is the most abundant fatty acid, followed by oleic acid (C18:1n9c) and stearic acid (C18:0). Compared to Group A3, Group A2 has higher levels of caprylic acid (C8:0), capric acid (C10:0), and lauric acid (C12:0). Group A4 exhibits higher concentrations of margaric acid (C17:0), caproic acid (C6:0), and tridecanoic acid (C13:0). Alpha-linolenic acid (C18:3n3) is lowest in Group A2 (*p* < 0.001), whereas tricosanoic acid (C21:0) is higher in Group A4. These results are consistent with previous studies on the fatty acid composition of yak milk [24,25].

In yak ghee, (Table 3), palmitic acid (C16:0) and oleic acid (C18:1n9c) are the most prevalent. The average contents of butyric acid (C4:0), caproic acid (C6:0), and caprylic acid (C8:0) are higher than those in cow ghee (3.79 g/100 g, 2.29 g/100 g and 1.39 g/100 g, respectively). These short-chain fatty acids can be rapidly absorbed and converted into energy; thus, consuming yak ghee may provide users with a readily available source of energy [26]. This may be due to smaller fat globules in yak milk compared to cow milk, offering a larger interfacial area for gastric lipase, which preferentially releases these fatty acids [27]. The composition of yak milk will directly affect the composition of yak ghee.

Conjugated linoleic acid (CLA) C9T11 was detected in Tibetan yak ghee at proportions of 1.57 g/100 g, 1.68 g/100 g, 1.62 g/100 g and 1.59 g/100 g in samples A1, A2, A3, and A4, respectively, surpassing the CLA content in cow milk ghee [28]. This elevated CLA level underscores the superior nutritional value of Tibetan yak ghee compared to standard ghee [16]. The content of palmitic acid (C16:0) and oleic acid (C18:1n9c) in Tibetan yak ghee is relatively high, which is related to the high content of linoleic acid (C18:2n6c) and palmitic acid (C16:0) in the yak’s pasture. Linoleic acid undergoes biohydrogenation in the yak’s rumen, resulting in the formation of palmitic acid (C16:0) [29]. These fatty acids enhance the antioxidant properties and storage stability of yak ghee [30,31].

### 3.2. Correlation Analysis of Fatty Acids in Pasture Grass–Yak Milk–Yak Ghee System across Different Altitudes

According to the results from correlation heatmaps (Figure 3), the proportion of saturated fatty acids (SFA) in pasture grass from different altitudinal regions shows correlations with both the saturated and unsaturated fatty acids in yak milk and yak ghee. There is a strong positive correlation between the proportion of UFA in pasture grass and the ratio of polyunsaturated to saturated fatty acids in yak milk. Additionally, the proportion of polyunsaturated fatty acids (PUFA) in pasture grass shows a strong positive correlation with the PUFA proportion in both yak milk and yak ghee. This is because the pasture grass, as the sole food source for yaks, provides a sustainable source of PUFA in the milk fat [32], confirming the superiority of grass-fed yak milk fat.

Fatty acid composition varied significantly across different altitudinal samples (*p* < 0.05). Linear discriminant analysis (LDA) based on altitude (Figure 4A) indicates that the first and second discriminant functions account for 75.76% and 17.72% of the variance. Together, they cover 93.48% of the total variance, indicating adequate separation of fatty acid data across four altitude levels (A1 to A4). Specifically, the first discriminant function distinguishes the groups with altitude gradients A3 and A4, and the second discriminant function distinguishes the groups with altitude gradients A1 and A2. The divisions based on altitude gradients are primarily attributed to environmental factors influencing the transfer between pasture grass, milk, and ghee. The fatty acids in the pasture grass are affected by temperature [33] and sunlight [34], and with increasing altitude, temperatures decrease while sunlight intensity increases. As shown in Figure 5, unsaturated fatty acids, which are predominant in pasture grass, significantly increase in proportion with altitude within a 1000 m gradient. This is because UFA maintains cell membrane fluidity and function, which is crucial for the survival of pasture grass in extreme environments [35]. The proportion of UFA in yak milk from summer pastures in the Qinghai–Tibet Plateau is similar to previous findings, and higher than those from winter pastures, due to the increased proportion of fresh pasture grass in the diet leading to an increase in unsaturated fatty acids in the milk [36].

MANOVA revealed significant differences in fatty acid composition among pasture grass, yak milk, and yak ghee (*p* < 0.05). The LDA plot (Figure 4B) shows that the first discriminant function explains 98.6% of the variance, separating yak milk from yak ghee, and the second function (1.4% variance) distinguishes pasture grass. This separation is due to microbial lipid synthesis in the rumen [37], which alters fatty acid profiles in milk and ghee [38]. Figure 4 illustrates distinct fatty acid profiles for pasture grass compared to yak milk and ghee. The fatty acid compositions of yak milk and ghee are similar, with minimal changes during ghee production [17,39]. However, yak ghee has reduced polyunsaturated fatty acids compared to yak milk, likely due to processing-induced oxidative degradation [13].

### 3.3. Correlation Analysis of Fatty Acid Indicators in Pasture Grass–Yak Milk–Yak Ghee System at the Same Altitude

Regression analysis was conducted on pasture grass, milk, and ghee from the same altitude, as shown in Table 4. The results for groups A4 and A3 exhibit better linear fitting for saturated fatty acids, unsaturated fatty acids, and especially polyunsaturated fatty acids compared to groups A2 and A1, which may be due to the harsh high-altitude environment prompting pasture grass and yaks to adjust their biosynthesis of lipids to adapt. The ruminal and metabolic mechanisms of Tibetan yaks on the Qinghai–Tibet Plateau significantly influence the dominant fatty acids in milk [29]. The R^2^ values for linear fits in the A4 group were 0.662, 0.587, and 0.784, respectively. The regression equations indicate a negative correlation between pasture grass fatty acid indicators and yak milk fatty acids, except for polyunsaturated fatty acids in group A3. Conversely, yak ghee shows a positive correlation with yak milk indicators. Linear fitting was less effective for groups A1 and A2, potentially due to a reduced impact of environmental factors on physiological processes in these regions or the influence of factors like soil type, water availability, and temperature fluctuations on fatty acid composition [40,41].

### 3.4. Functional Fatty Acids in Pasture Grass–Yak Milk–Yak Ghee System

Functional fatty acids, particularly ω-3 fatty acids, are present in varying concentrations in pasture grass, yak milk, and yak ghee. Notably, α-linolenic acid (C18:3n3) is found in high levels in pasture grass (Figure 6) and shows a significant increase (*p* < 0.001) along altitude gradients from A1 to A2 and from A3 to A4. Conversely, α-linolenic acid levels are markedly lower in yak milk and yak ghee (*p* < 0.001). The concentrations of eicosatetraenoic acid (EPA) and docosahexaenoic acid (DHA) in yak ghee are significantly higher than those in pasture grass (*p* < 0.001), likely due to the in vivo conversion of α-linolenic acid into EPA and DHA [42]. The levels of EPA and DHA in pasture grass are notably lower at the A2 altitude compared to A1, A3, and A4 (*p* < 0.001). In contrast, EPA content in yak ghee is significantly higher at A2 compared to A1, A3, and A4 (*p* < 0.001). DHA content peaks in pasture grass at the A2 altitude and declines at higher altitudes, whereas in yak ghee, DHA content peaks at A3 altitude. γ-linolenic acid (C18:3n6) was not detected in yak milk and ghee but is abundant in pasture grass, particularly at A2 altitude. Research suggests that γ-linolenic acid (C18:3n6) synthesis pathways are upregulated in yak serum at high altitudes [43]. As γ-linolenic acid (C18:3n6) contributes to cell membrane phospholipids and aids in protection against inflammation and oxidative stress, its increased availability at high altitudes may support yaks’ adaptation to harsh conditions [43]. Consequently, the reduced levels of γ-linolenic acid (C18:3n6) in yak milk and ghee may reflect its greater utilization in cell membrane construction. Linoleic acid (C18:2n6c) was present in lower concentrations in yak milk and ghee (Figure 6). The correlation observed in Figure 4 indicates that as the linoleic acid (C18:2n6c) content in forage increases, so does the content of PUFA in yak milk. Linoleic acid (C18:2n6c) likely serves as a precursor for CLA and ω-3. In yaks, linoleic acid (C18:2n6c) is preferentially used for PUFA synthesis, thus enhancing the PUFA content in milk fat [44]. Oleic acid (C18:1n9c) is less prevalent in pasture grass but reaches its peak concentration in yak milk and ghee at A3 and A1 altitudes, respectively. The higher C18:1n9c levels in yak milk and yak ghee may result from its conversion from palmitic acid rather than direct dietary intake from pasture grass [45]. C18:1n9c is also associated with calf growth and development, leading to its incorporation into milk fat during yak milk synthesis [46].

## 4. Conclusions

This study investigates the effect of altitude on the fatty acid composition of the pasture grass–yak milk–yak ghee system on the Qinghai–Tibet Plateau. The results show that beneficial unsaturated fatty acids, such as oleic acid (C18:1) and linoleic acid (C18:2n6c), are more prevalent at higher altitudes. These findings indicate that altitude significantly influences yak lipid metabolism and the nutritional quality of dairy products. This adaptation benefits high-altitude residents by helping them cope with the harsh conditions of the plateau. Furthermore, this research enhances our understanding of fatty acid distribution in yak dairy products and offers insights for optimizing dairy production in high-altitude environments.

## Figures and Tables

**Figure 1 animals-14-02975-f001:**
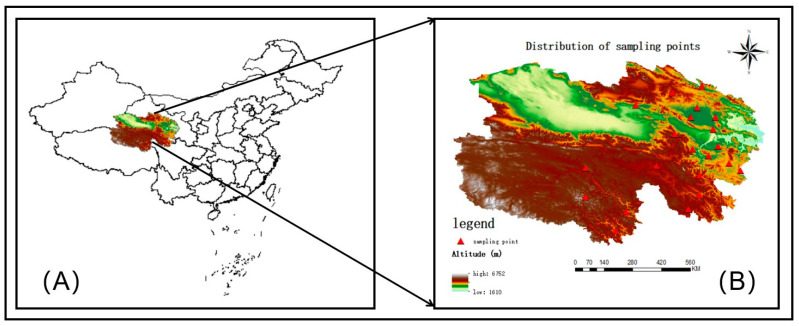
Distribution of sampling points for pasture grass, yak milk, and yak ghee sample collection. (**A**) Map of China; (**B**) Distribution of sampling points in Qinghai Province.

**Figure 2 animals-14-02975-f002:**
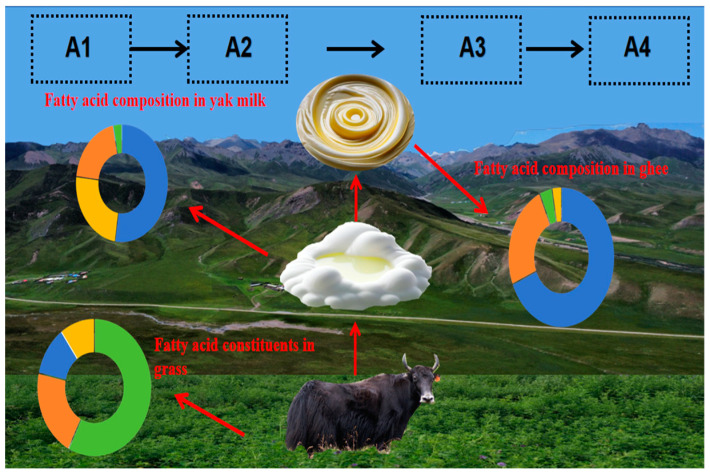
Diagram of the overall idea of the experiment (the blue area represents the proportion of saturated fatty acids, the orange area represents the proportion of unsaturated fatty acids, the yellow area represents the proportion of polyunsaturated fatty acids, and the green area represents the proportion of polyunsaturated fatty acids to saturated fatty acids).

**Figure 3 animals-14-02975-f003:**
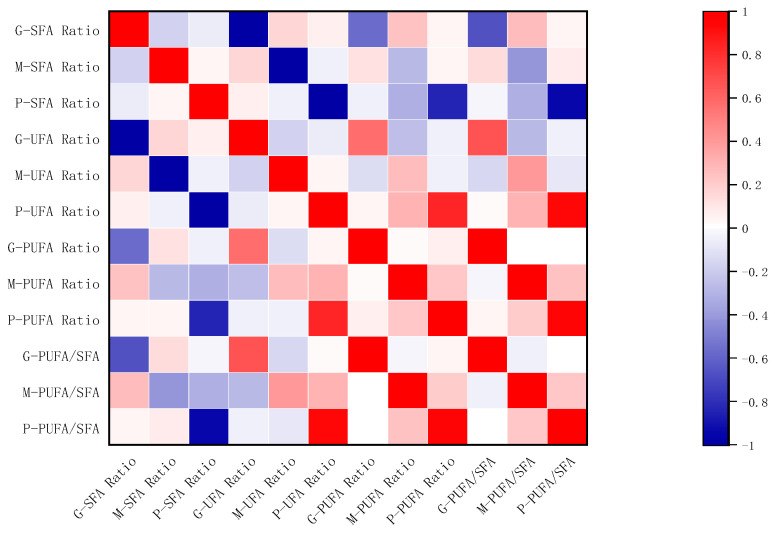
Correlation heat map of four fatty acids in pasture grass, yak milk, and yak ghee. P, pasture grass; M, yak milk; G, yak ghee (red indicates a positive correlation, blue indicates a negative correlation, and the darker the color, the stronger the correlation to the corresponding indicator).

**Figure 4 animals-14-02975-f004:**
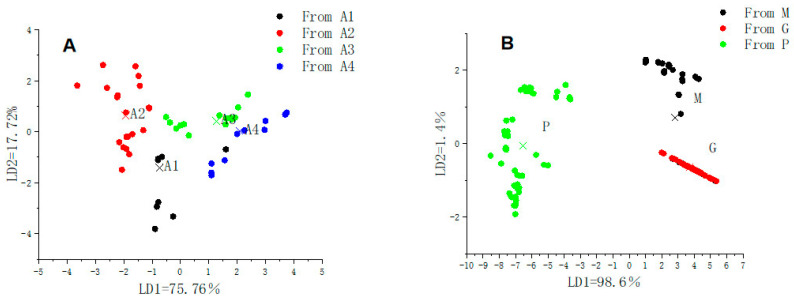
(**A**) is an LDA plot based on the altitudinal gradient, (**B**) is an LDA plot based on the type of sample, P, pasture grass; M, yak milk; G, yak ghee. A1 was grazed on rangeland at an altitude of about 3100 m; A2 was grazed on rangeland at an altitude of about 3600 m; A3 was grazed on rangeland at an altitude of about 4100 m; A4 was grazed on rangeland at an altitude of about 4600 m.

**Figure 5 animals-14-02975-f005:**
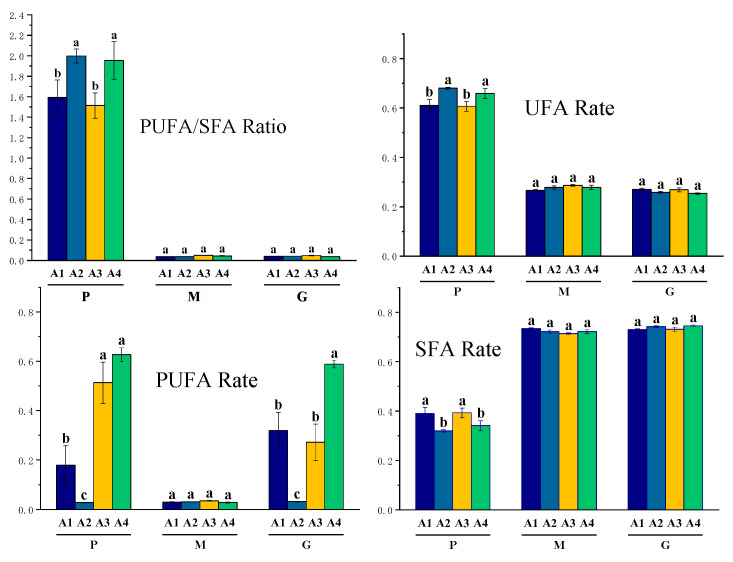
Multifactorial bar chart of multivariate analysis for pasture grass-yak milk-yak ghee fatty acids, P, pasture grass; M, yak milk; G, yak ghee. A1 was grazed on rangeland at an altitude of about 3100 m; A2 was grazed on rangeland at an altitude of about 3600 m; A3 was grazed on rangeland at an altitude of about 4100 m; A4 was grazed on rangeland at an altitude of about 4600 m. Different letters on the bars indicate significant differences *p* < 0.05.

**Figure 6 animals-14-02975-f006:**
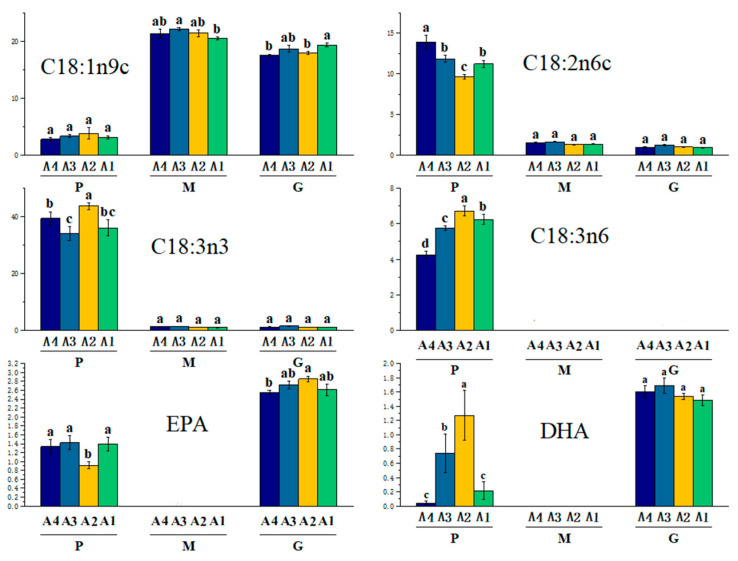
Multifactor bar chart of functional fatty acids in pasture grass-yak milk-yak ghee, M, yak milk; P, pasture grass; G, yak ghee. A1 was grazed on rangeland at an altitude of about 3100 m; A2 was grazed on rangeland at an altitude of about 3600 m; A3 was grazed on rangeland at an altitude of about 4100 m; A4 was grazed on rangeland at an altitude of about 4600 m. Different letters on the bars indicate significant differences (*p* < 0.05).

**Table 1 animals-14-02975-t001:** Fatty acid content in pasture grass (g/100 g) samples collected from four altitudes in Qinghai Province.

	Altitude Gradient ^1^						
Fatty Acids	A1	A2	A3	A4	Mean	SEM	*p*-Value
C4:0	0.20	0.21	0.21	ND	0.207	0.016	0.975
C6:0	0.44	0.49	0.55	0.50	0.495	0.038	0.878
C8:0	0.26	0.30	0.31	0.30	0.293	0.021	0.095
C10:0	0.19	0.09	ND ^2^	ND	0.140	0.050	0.051
C11:0	0.90	0.72	0.67	0.79	0.770	0.077	0.798
C12:0	0.77 ^b^	1.44 ^a^	0.59 ^b^	0.39 ^b^	0.798	0.091	<0.001
C13:0	0.50	0.46	0.84	0.57	0.593	0.067	0.448
C14:0	1.15 ^b^	1.86 ^a^	1.02 ^b^	1.11 ^b^	1.285	0.075	<0.001
C15:0	0.90	1.06	0.96	0.97	0.973	0.120	0.975
C16:0	20.60	21.31	19.93	20.69	20.63	0.443	0.731
C16:1	0.43	0.40	0.38	0.43	0.410	0.018	0.746
C17:0	0.37	0.59	0.36	0.30	0.405	0.052	0.231
C17:1	0.27	1.23	0.22	0.32	0.510	0.169	0.108
C18:0	4.83 ^a^	5.40 ^a^	5.13 ^a^	3.36 ^b^	4.680	0.166	<0.001
C18:1n9t	0.22	0.30	0.27	0.37	0.290	0.022	0.096
C18:1n9c	4.03	4.03	2.94	2.84	3.460	0.296	0.301
C18:2n6c	13.34 ^a^	10.26 ^b^	11.02 ^b^	13.86 ^a^	12.12	0.333	<0.001
C20:0	1.52 ^ab^	1.25 ^b^	1.90 ^a^	2.01 ^a^	1.670	0.098	0.016
C18:3n6	5.25 ^bc^	5.92 ^ab^	6.28 ^a^	4.56 ^c^	5.503	0.157	<0.001
C18:3n3	33.89	35.99	40.00	37.66	36.88	1.140	0.280
C21:0	0.22 ^b^	0.37 ^a^	0.29 ^ab^	0.25 ^ab^	0.283	0.024	0.050
C20:2	2.39 ^ab^	2.64 ^ab^	2.28 ^b^	3.40 ^a^	2.678	0.118	<0.001
C22:0	1.75 ^ab^	1.36 ^b^	1.99 ^ab^	2.24 ^a^	1.835	0.022	0.024
C24:0	4.69	4.82	3.58	4.01	4.275	0.362	0.071
C20:5n3	1.37 ^ab^	1.11 ^b^	1.11 ^b^	1.50 ^a^	1.273	0.602	0.032
C22:6n3	0.64 ^b^	0.79 ^b^	2.04 ^a^	0.25 ^b^	0.930	0.196	0.002

^1^ Altitude gradient: A1 was grazed on rangeland at an altitude of about 3100 m; A2 was grazed on rangeland at an altitude of about 3600 m; A3 was grazed on rangeland at an altitude of about 4100 m; A4 was grazed on rangeland at an altitude of about 4600 m, the means with different superscripts within the same row bearing different superscripts are significantly different (*p* < 0.05). ^2^ ND, not detected; Mean, arithmetic mean; SEM, standard error of the mean. ^a,b,c^ Different letters within different altitude groups indicate significant differences (*p* < 0.05).

**Table 2 animals-14-02975-t002:** Fatty acid content in yak milk (g/100 g) samples collected from four altitudes in Qinghai Province.

	Altitude Gradient ^1^						
Fatty Acids	A1	A2	A3	A4	Mean	SEM	*p*-Value
C4:0	1.12	1.11	1.03	1.14	1.100	0.179	0.131
C6:0	1.84	1.82	1.71	1.86	1.808	0.026	0.167
C8:0	1.16	1.93	1.10	1.21	1.350	0.023	0.303
C10:0	2.36	2.43	2.18	2.39	2.340	0.048	0.258
C12:0	1.80 ^ab^	1.96 ^a^	1.74 ^b^	1.89 ^ab^	1.848	0.035	0.050
C13:0	0.08 ^b^	0.08 ^b^	0.08 ^b^	0.09 ^a^	0.083	0.001	0.004
C14:0	9.02 ^ab^	9.60 ^a^	8.78 ^b^	9.28 ^ab^	9.170	0.111	0.038
C14:1	0.36 ^b^	0.42 ^a^	0.37 ^b^	0.39 ^ab^	0.385	0.008	0.020
C15:1	1.57 ^a^	1.57 ^a^	1.52 ^ab^	1.45 ^b^	1.528	0.019	0.050
C16:0	34.51 ^ab^	34.78 ^a^	32.71 ^b^	33.67 ^ab^	33.918	0.330	0.978
C16:1	2.24 ^b^	2.47 ^a^	2.23 ^b^	2.22 ^b^	2.290	0.024	0.009
C17:0	1.02	1.08	0.99	1.10	1.048	0.027	<0.001
C17:1	0.37 ^ab^	0.46 ^a^	0.36 ^b^	0.38 ^ab^	0.393	0.016	0.037
C18:0	17.65 ^ab^	16.40 ^b^	18.71 ^a^	17.48 ^ab^	17.560	0.296	0.035
C18:1n9t	0.28 ^ab^	0.25 ^b^	0.30 ^a^	0.27 ^ab^	0.275	0.007	0.049
C18:1n9c	21.00	21.2	22.17	21.34	21.428	0.250	0.480
C18:2n6t	0.29 ^c^	0.24 ^b^	0.29 ^b^	0.35 ^a^	0.293	0.008	0.010
C18:2n6c	1.41 ^a^	1.19 ^b^	1.58 ^a^	1.48 ^ab^	1.415	0.033	0.011
C20:0	0.37	0.40	0.39	0.36	0.380	0.007	0.272
C20:1	0.19 ^ab^	0.23 ^a^	0.23 ^a^	0.14 ^b^	0.198	0.026	0.030
C18:3n3	1.31 ^a^	0.98 ^b^	1.52 ^a^	1.38 ^a^	1.298	0.097	<0.001
C21:0	0.16	0.11	0.13	0.17	0.143	0.010	0.101
C22:0	0.15	0.18	0.15	0.16	0.160	0.006	0.325
C20:4n6	0.17	0.17	0.16	0.13	0.158	0.009	0.669
C24:0	ND ^2^	0.12	0.10	ND	0.110	0.010	0.192

^1^ Altitude gradient: A1 was grazed on rangeland at an altitude of about 3100 m; A2 was grazed on rangeland at an altitude of about 3600 m; A3 was grazed on rangeland at an altitude of about 4100 m; A4 was grazed on rangeland at an altitude of about 4600 m, the means with different superscripts within the same row are significantly different (*p* < 0.05). ^2^ ND, not detected; Mean, arithmetic mean; SEM, standard error of the mean. ^a,b,c^ Different letters within different altitude groups indicate significant differences (*p* < 0.05).

**Table 3 animals-14-02975-t003:** Fatty acid content in yak ghee (g/100 g) samples collected from four altitudes in Qinghai Province.

	Altitude Gradient ^1^						
Fatty Acids	A1	A2	A3	A4	Mean	SEM	*p*-Value
C4:0	3.67 ^ab^	3.46 ^b^	4.04 ^ab^	4.63 ^a^	3.950	0.183	<0.032
C6:0	4.99	3.36	3.64	4.41	4.100	0.335	0.852
C8:0	1.68	1.55	1.65	1.95	1.708	0.073	0.551
C10:0	3.01	2.71	2.87	3.36	2.988	0.118	0.596
C11:0	0.025 ^a^	0.020 ^b^	0.022 ^ab^	0.025 ^a^	0.023	0.001	0.005
C12:0	2.22	1.95	2.11	2.24	2.130	0.069	0.816
C14:1	0.932	0.942	1.106	0.900	0.970	0.054	0.615
C14:0	9.46	9.05	9.10	9.32	9.233	0.138	0.944
C15:0	1.37 ^ab^	1.43 ^a^	1.41 ^ab^	1.32 ^b^	1.383	0.017	0.050
C16:1	1.38	1.35	1.35	1.41	1.373	0.021	0.499
C16:0	30.44	30.37	29.42	29.59	29.95	0.388	0.070
C18:0	14.06	14.40	14.03	12.83	13.83	0.265	0.199
T9C18:1	0.39	0.41	0.40	0.37	0.393	0.016	0.348
C9C18:1	18.77 ^ab^	19.07 ^a^	18.79 ^ab^	17.67 ^b^	18.57	0.229	0.021
T6C18:2	0.30 ^ab^	0.33 ^a^	0.27 ^b^	0.29 ^ab^	0.298	0.009	0.035
C6C18:2	1.02 ^b^	0.88 ^b^	1.20 ^a^	1.00 ^b^	1.025	0.034	<0.001
C20:0	0.30	0.36	0.33	0.29	0.320	0.011	0.106
N3C18:3	1.28 ^b^	1.20 ^b^	1.55 ^a^	1.27 ^b^	1.325	0.039	0.022
C21:0	1.11 ^a^	0.10 ^ab^	0.09 ^ab^	0.08 ^b^	0.345	0.004	0.049
C20:2	0.07 ^ab^	0.08 ^a^	0.07 ^ab^	0.06 ^b^	0.070	0.018	0.005
C8C20:3	0.17	0.10	0.10	0.08	0.113	0.005	0.066
C22:0	0.005 ^b^	0.016 ^b^	0.047 ^a^	0.003 ^b^	0.018	0.001	0.042
C22:1	0.006 ^b^	0.016 ^a^	0.017 ^a^	0.011 ^ab^	0.013	0.021	0.028
C20:4	0.10	0.16	0.09	0.07	0.105	0.003	0.138
C23:0	0.08	0.10	0.09	0.08	0.088	0.002	0.148
C20:5	0.070 ^b^	0.067 ^a^	0.068 ^ab^	0.059 ^ab^	0.066	0.025	0.033
C24:0	0.095 ^ab^	0.189 ^a^	0.106 ^ab^	0.103 ^ab^	0.123	0.002	0.022
C24:1	0.018 ^a^	0.020 ^a^	0.013 ^ab^	0.008 ^ab^	0.015	0.002	0.006
C22:6	0.030 ^a^	0.025 ^a^	0.025 ^a^	0.015 ^b^	0.024	0.002	0.041
anteisoC13:0	0.063	0.061	0.069	0.069	0.066	0.001	0.148
isoC13:0	0.022	0.026	0.023	0.023	0.024	0.021	0.730
C13:0	0.080	0.077	0.168	0.080	0.101	0.021	0.085
isoC14:0	0.22	0.25	0.26	0.25	0.245	0.006	0.983
anteisoC15:0	0.49 ^ab^	0.52 ^a^	0.49 ^ab^	0.45 ^b^	0.488	0.011	0.041
isoC15:0	0.72 ^b^	0.80 ^ab^	0.81 ^a^	0.77 ^ab^	0.775	0.014	0.050
isoC16:0	0.37	0.39	0.39	0.36	0.378	0.008	0.240
anteisoC17:0	0.35 ^ab^	0.37 ^a^	0.32 ^b^	0.27 ^c^	0.328	0.010	0.030
isoC17:0	0.50 ^ab^	0.53 ^a^	0.50 ^ab^	0.45 ^b^	0.495	0.012	0.012
C17:0	0.74	0.81	0.74	0.80	0.773	0.027	0.275
C9C17:1	0.34	0.34	0.34	0.37	0.348	0.012	0.813
C9T11CLA	1.57	1.68	1.62	1.59	1.615	0.052	0.415

^1^ Altitude gradient: A1 was grazed on rangeland at an altitude of about 3100 m; A2 was grazed on rangeland at an altitude of about 3600 m; A3 was grazed on rangeland at an altitude of about 4100 m; A4 was grazed on rangeland at an altitude of about 4600 m. The means with different superscripts within the same row are significantly different (*p* < 0.05). Mean, arithmetic mean; SEM, standard error of the mean. ^a,b,c^ Different letters within different altitude groups indicate significant differences (*p* < 0.05).

**Table 4 animals-14-02975-t004:** Regression analysis statistic and prediction equation of pasture grass, milk, and ghee at the same altitude.

Altitude	Content	R^2^	*p*-Value	Regression Equation
A4	SFA Rate	0.662	0.008	M-SFA Rate = −0.92 + 2.235G − SFA Rate − 0.075P − SFA Rate
	UFA Rate	0.587	0.008	M-UFA Rate = −0.239 + 2.235G − UFA Rate − 0.075P − UFA Rate
	PUFA Rate	0.784	0.000	M-PUFA Rate = 0.615 − 13.856G − PUFA Rate − 0.684P − PUFA Rate
A3	SFA Rate	0.647	0.004	M-SFA Rate = 0.54 + 0.154G − SFA Rate − 0.155P − SFA Rate
	UFA Rate	0.647	0.004	M-UFA Rate = 0.151 + 0.154G − UFA Rate − 0.155P − UFA Rate
	PUFA Rate	0.739	0.001	M-PUFA Rate = 1.536 − 37.874G − PUFA Rate + 1.076P − PUFA Rate
A2	SFA Rate	0.225	0.112	-
	UFA Rate	0.225	0.112	-
	PUFA Rate	0.095	0.259	-
A1	SFA Rate	0.129	0.161	-
	UFA Rate	0.129	0.161	-
	PUFA Rate	0.315	0.034	-

M, yak milk; P, pasture grass; G, yak ghee; A1 was grazed on rangeland at an altitude of about 3100 m; A2 was grazed on rangeland at an altitude of about 3600 m; A3 was grazed on rangeland at an altitude of about 4100 m; A4 was grazed on rangeland at an altitude of about 4600 m. SFA Rate, saturated fatty acids as a percentage of all fatty acids. UFA Rate, unsaturated fatty acids as a percentage of all fatty acids. PUFA Rate, polyunsaturated fatty acids as a percentage of all fatty acids. R^2^ represents the goodness of fit of a regression equation; the higher the R^2^, the better the model fits the data. *p*-value, a measure of the statistical significance of the effect of an independent variable on a dependent variable.

## Data Availability

Data openly available in a public repository.

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
