# Peer review of "Fatty Acids Composition of Pasture Grass, Yak Milk and Yak Ghee from the Four Altitudes of Qinghai–Tibet Plateau: A Predictive Modelling Approach to Evaluate the Correlation among Altitude, Pasture Grass, Yak Milk and Yak Ghee"

_animals, 2024, doi:10.3390/ani14202975_

Round 1
Reviewer 1 Report
Comments and Suggestions for Authors
This study investigates how the fatty acid composition of grass, yak milk, and yak ghee changes at different altitudes on the Qinghai-Tibet Plateau. Results will help improve dairy practices, ultimately contributing to better nutrition and food security for residents in high-altitude regions. Overall, the results were interesting, however, some problems must be addressed.
General issues
1. Specific heights at elevations A1-A4 are labeled in detail in Materials and Methods, and it is recommended that they be expressed as heights in the abstract.
2. Please clearly mark the number of biological replicates used for each part of the results.
3. Line 69 It is inappropriate to mention that yak ghee contains high levels of saturated fatty acids while mentioning high levels of unsaturated fatty acids.
4. Are the fatty acid determinations for pasture consistent with those for milk, please list them in the material methods.
5. Missing significance values for Table 3
6. Changes in fatty acid levels in pasture or yak milk should be reflected in the description of result 3.1. Line 143: In grass, lauric acid (C12:0) levels were significantly elevated……
7. The significance P should be italicized.
8. Should significance (p < 0.001) be indicated by capital letters? Significance levels were inconsistent in the methods and results.
9. Fatty acid composition as influenced by rumen microorganisms in the conclusion is not relevant to this study and is not recommended to be mentioned in the conclusion. In addition, the conclusions should be further simplified.
Small issues
1. 3.1 Fatty acid profiles in Grass-Yak Milk-Ghee
2. The three-line format of Table 1 and 2 is inappropriate.
3. Average should be replaced with mean.
Reviewer 2 Report
Comments and Suggestions for Authors
The animals-3212960 study examines the fatty acid composition of grass, yak milk, and yak ghee at four different altitudes on the Qinghai-Tibet Plateau. It uses predictive modeling to explore the correlations between altitude and grass, yak milk, and yak ghee composition. The article is well-structured, but some points need to be addressed.
1. The grammar and punctuation have to be thoroughly revised throughout the manuscript. In many places, commas, semicolons/colons are either missing or ignored. Authors are requested to carefully check these.
Line 49 (low-oxygen environments[1] .), 51 (inhabitants[2-3].), 53 (milk. Additionally. the yak), 63, 65, line 118, 137, 152, 154, 156, 178, 206 and many others.
2. Abbreviations/Acronyms must be spelled out and explained when first introduced in the text.
Line 69 SFA (SATURATES FATTY ACIDS), line 73,
3. I recommend revising lines 58-60, 101-102 and 134 for clarity, as they are currently difficult to understand.
4. The introduction should include an overview of the process and technology used to obtain yak ghee.
5. In the material and method section:
- When referring to dry ice, additional details should be provided, such as its composition (solid carbon dioxide), its temperature, and its common uses for cooling and preservation.
- The authors should provide a clearer and more detailed explanation of their sample collection method. It is known that the fatty acid composition of milk changes throughout the milking process (see also "Within-Milking Variation in Milk Composition and Fatty Acid Profile of Holstein Dairy Cows").
- To what temperature were the samples cooled, given that the outside temperature was 10°C? (line 106)
- Figure 1 could be improved by adding information on the altitudes.
- Line 111 Additional information should be included.
- The cited source may be incorrect or confusing, as Luo et al. describe a different methodology.
- The procedure for identifying fatty acids should be detailed, including the method used to calculate the concentration of each fatty acid in the samples.
- Furthermore, clarify the reference standard employed for quantifying the fatty acid concentrations in the samples.
- The methodology for determining the fatty acid profile of the collected plant material should be described in detail, including the specific analytical techniques and procedures used for this analysis.
- The titles of Tables 2 and 3 are too lengthy, and some explanations are missing, including the meanings of the small letters and acronyms used.
- Figure 3 requires additional explanation to clarify its content and significance.
6. I recommend a major revision.
Reviewer 3 Report
Comments and Suggestions for Authors
I cannot recommend your paper for publication in its current form. It is not well presented. Many of your statements do not follow from the data in the tables. The regression equations are not clearly presented. You have no discussion of the possible confounding of different varieties of grass species associated with the change in altitude. The links to human health with the changes in fatty acids in milk and ghee are a stretch. How would you change the profiles without more closely characterizing the grass content of the pastures. You assume the changes are due to altitude. Is it due to the grass types or due to animal physiology or both? See specific comments below.
Title: Fatty acids composition of grass, yak milk and yak ghee from the four altitudes of Qinghai-Tibet Plateau: A predictive modelling approach to evaluate the correlation among altitude, grass, yak milk and yak ghee.
Authors:
Runze Wang, Jinfen Yang, Binqiang Bai, Muhammad Irfan Malik, Yayu Huang, Yingkui Yang, Shujie Liu,
Xuefeng Han, Lizhuang Hao.
Aim of study:How fatty acid composition of grass, yak milk, and yak ghee changes at different altitudes. How altitude influences the nutritional value of these products.
Table 1. There is a superscript on C18:3n6 of "2" but no footnote in the table for what it refers to. There should be a superscript on Altitude Gradient to refer to the legend of the table for the altitudes for A1, A2, A3 and A4. There is superscript of 1 on the ND for C10:0 under A3 with no explanation.
Lines 173-174 Group A2 had higher levels of caprylic acid, capric acid and lauric acid compared to group A3… The statistics indicate that caprylic and capric are not different across altitudes. In fact the statistical discussion from 171 to 181 is misleading. Alpha-linoleic acid is not different in group A3 and A4 and A1. Numberically it was greater, but not significantly. The important observation is A2 had lower alpha-linolenic acid compared to the other three elevations.
Line 183 Butyric acid is greatest in A4, but it is not different from A1 and A3, the superscripts are not different. A2 has the lowest concentration of butyric acid.
Line 184-186 The rise in butyric with altitude is not clear, as see above. In addition, the total fatty acid content of the ghee would determine the energy value, and not the butyric acid alone. In fact, the proportions of fatty acids in the ghee may actually have a lower energy value as altitude increases.
Line 187 - 190 Short and medium chains (C4 to C15) are 26.3 to 30.1% of the fatty acids in ghee. They are not the major fatty acid components.
Line 192-193 The C9T11 reported in table 3 is 1.57, 1.68, 1.62, 1.59 g/100. These are percents. Where are you getting the numbers you report? Possibly
Line 204 Biohydrogenation of C18:2n6c would no yield palmitic acid. Palmitic acid is high in the grass samples.
Figure 3. Needs a better legend to explain the items on the map. A better explanation of how the ratios are calculated in needed. Possibly simplify to the main fatty acid relationships you are interested in.
Lines 228 to 236 Interesting discussion of sunlight and altitude. In addition,, does the CO2 concentration change with altitude or become less concentrated which could influence photosynthesis.
Figure 5. Other than for PUFA, there is not a clear directionality with elevation. How might the variety of grass species confound the influence of elevation you are seeking to claify.
Table 4. It is very difficult to distinguish what is in the regression equations. Suggest you change your format. Why not do a regression with altitude was a independent variable? Very confusing table.
Table 4 has a column labeled content, e.g. SFA Rate, then the equation is
M-SFA = -0.92 + 2.23*G-SFA - 0.075*P-SFA
I assume G-SFA indicates grass, does P-SFA also indicate grass (pasture)? It would not make sense to be polyunsaturated SFA.
Line 258 - 262 This statement is difficult to confirm with how the data is presented in table 4.
Lines 267-270 This conclusion needs better clarification and support.
Could be improved.
Round 2
Reviewer 2 Report
Comments and Suggestions for Authors
Dear Authors,
Thank you for the corrections made. The manuscript has been significantly improved.
Best regards
Author Response
Dear Editor,
Thank you for your support. It was your guidance that helped me improve this manuscript.
Best regards!